# Diabetes management cascade in Ulaanbaatar, Mongolia

Orkhonselenge Davaadamdin[1], Bolor-Erdene Ganbat[1], Uurtsaikh Baatarsuren[1,2], Khulan Tuvdendarjaa[1,2], Maral Myanganbayar[1], Delgerbat Boldbaatar[1,3], Yanfang Su[4], Andreas Bungert[1], Myagmartseren Dashtseren[2,5], Tsolmon Unurjargal[2,5], Johannes Boch[6], Naranbaatar Dashdorj[1], Naranjargal Dashdorj[1,3,2]*

1 Onom foundation, Ulaanbaatar, Mongolia, 2 Mongolian Society of Hypertension, Ulaanbaatar, Mongolia, 3 The Liver center, Ulaanbaatar, Mongolia, 4 Department of Global Health, University of Washington, Seattle, Washington, United States of America, 5 Mongolian National University of Medical Sciences, Ulaanbaatar, Mongolia, 6 Novartis Foundation, Basel, Switzerland

* dashdorj@onomfoundation.org

## Abstract

### Objectives

This study aims to determine the prevalence of diabetes and impaired glucose tolerance among the adult population in Ulaanbaatar, identify gaps in clinical care by assessing awareness, treatment, and control of the disease, and examine the trends in diabetes risk factors.

### Methods

A representative sample of 2,169 adults aged 20 years and older from 9 districts of Ulaanbaatar was randomly selected for a cross-sectional survey conducted in 2018. Physical measurements were taken, and data were collected using the Mongolian version of the World Health Organization Instrument for Non-communicable Disease Risk Factor Surveillance (STEPS).

### Results

The overall prevalence of diabetes among adults aged 20 and above was 6.27% (n = 136). Diabetes prevalence was significantly higher in men 7.5% (n = 73), than in women (5.3% (n = 63). OR=1.63, P < .007).
Among those with diabetes, 48.5% were unaware of their condition, 15.4% were diagnosed but untreated, 30.2% were treated but uncontrolled, and only 5.9% achieved glycemic control.

### Discussion

A major challenge in diabetes control in Ulaanbaatar is the low awareness and poor care-seeking behavior, especially among younger men. This reflects systematic and

provided the original author and source are credited.

**Data availability statement:** All data underlying the results presented in this study are fully available without restriction. The minimal anonymized dataset necessary to replicate the study findings has been deposited in the Zenodo repository and is accessible at: https://doi.org/10.5281/zenodo.17766598.

**Funding:** The author(s) received no specific funding for this work.

**Competing interests:** The authors have declared that no competing interests exist.

social challenges surrounding diabetes management in Mongolia. Further research is needed to explore barriers and opportunities to improve awareness and care.

## Introduction

Non-communicable diseases (NCDs) have become an enormous health burden in low- and middle-income countries (LMICs), including Mongolia. It poses a threat to economic and social development, as well as population health. Cardiovascular diseases (CVDs) are the top cause of mortality in Mongolia, and hypertension (HTN), coupled with diabetes, is the most significant risk factor of cardiovascular disease [1]. Hemorrhagic stroke cases registered in Mongolia comprise almost 80% of all hemorrhagic stroke cases worldwide, making Mongolia one of the leading countries with the highest stroke-related mortality rates in the world [2].

The International Diabetes Federation (IDF) reports that, as of 2021, approximately 537 million (10.5%) adults (20–79 years) worldwide live with diabetes; by 2045, this number will increase to 783 million (12.2%). Another 10% of adults (541 million) have impaired glucose tolerance (IGT), and this number is projected to reach 730 million (11.4%) by 2045. The IDF Western Pacific Region is home to 38% of all people living with diabetes worldwide. Diabetes was responsible for 717,400 deaths in the IDF Western Pacific Region in 2019 — the highest number of deaths due to diabetes among all IDF regions. One in two adults with diabetes are undiagnosed, and 88% of undiagnosed diabetics live in LMICs. Diabetes prevalence in LMICs is increasing faster than in high-income countries (HICs), yet awareness, treatment, and control of diabetes in developing countries remain much lower [3].

The prevalence of diabetes in Mongolia was most recently assessed in 2019 using the World Health Organization (WHO) stepwise approach to non-communicable diseases surveillance (STEPS). In this survey, diabetes was defined as a fasting plasme glucose level ≥7.0 mmol/L, or previous diagnosis of diabetes by healthcare professionals. Pre-diabetes, or impaired fasting glucose (IFG), was defined as a fasting plasma glucose level between 6.1 and 6.9 mmol/L. Based on these definitions, the survey found diabetes prevalence of 9.5% overall (8.9% in urban areas and 7.7% in rural areas), and pre-diabetes prevalence of 17.4% (18.7% in urban areas and 16.6% in rural areas) [4]. Non-declining NCD risk factors, as well as poor knowledge of diabetes among the general population greatly contribute to the development of the disease. Demaio et. al. analyzed data from the national NCD Knowledge, although the survey was conducted in Mongolia in 2010, it remains the most comprehensive source of data on public awareness of diabetes in Mongolia and serves as the best available indicator of awareness levels at that time [5].

Being unaware, untreated, and uncontrolled can lead to dire consequences. In Mongolia, a slight complication of diabetes can result in a severe negative impact on patients in a short period. A cross-sectional study conducted by Sainbileg S. et al. indicated poor glycemic control among diabetic patients: HbA1C and fasting blood glucose levels were at 8.92±0.71% and 9.47±2.49 mmol/l, respectively. Mean

diabetes duration was 7.23 ± 3.08 years, and the prevalence of diabetic retinopathy, nephropathy, peripheral neuropathy was at 29.3%, 14.8%, and 71.0%, respectively [6]. The International Diabetes Federation (IDF) reported that health costs of treating the complications of diabetes account for over 50% of the direct health costs of diabetes [7].

Mongolia is a nation undergoing a rapid epidemiological transition and urbanization, a process that will continue in the decades to come and likely further drive the burden of diabetes. This study was conducted within the framework of Better Hearts Better Cities, following the CARDIO4Cities approach that aims to reduce the burden of CVD and its risk factors such as hypertension by strengthening CVD risk management in urban centers, including Ulaanbaatar, the most populous city in Mongolia, with over 1.6 million people representing 48.1% of the national population [8,9]. Diabetes is a major comorbidity of hypertension and influences management of CVD outcomes. Overall, 1 mmol/l change to the lower than usual fasting glucose was associated with a 21% (95% CI 18–24%) lower risk of total stroke and a 23% (19–27%) lower risk of total IHD [10].

Here, we report the prevalence of diabetes mellitus and gaps in clinical care (i.e., lack of awareness of diagnosis, lack of treatment of the diagnosed, and lack of control among the treated) in Ulaanbaatar. Although previous studies have analyzed the prevalence of diabetes in Mongolia, the results of those studies are not specific to Ulaanbaatar [4]. The study at hand examines the prevalence of diabetes and impaired glucose tolerance to have an extended view of diabetes and its risk factor trends in the city. Furthermore, we investigated the awareness, treatment, and control levels among diabetics to obtain the baseline overview of diabetes care.

## Methods

### Study design and population

A two-stage cluster sampling was used to select adults aged 20 years and above living in Ulaanbaatar city, first by sampling administrative units (khoroo), and then by sampling individuals within these units. Guided by the WHO STEPS Surveillance Manual, a sample size of 4511 was required for the total population for the STEPS Ulaanbaatar survey with a confidence level of 1.96 (associated with a 95% confidence interval), a margin of error of 0.05, a design effect of 1.50, and an anticipated response rate of 80% [11]. For the laboratory analysis, every second participant aged 20 years and above was systematically selected and recruited, resulting in a final sample of 2169 individuals.

### Sampling

The WHO STEPS Surveillance Manual recommends that at least 50 primary sampling units (PSUs) are selected from over 100. In Ulaanbaatar, there are 142 primary health care centers (PHCCs) providing healthcare to all citizens. Every citizen is registered at their local PHCC, and thus PHCCs were appropriate to serve as PSUs to sample the general population. One district, Bagakhangai, was excluded because it was geographically isolated and had a small population. Further, the proportional probability sampling was used to select 52 PHCCs from the remaining eight districts in the first stage of cluster sampling. In the second stage, 88 individuals aged 20 years or older were randomly selected from the registries of each of the 52 PHCCs. If participants could not be reached by the research team, they were replaced by the next participant within the same age and sex category.

### Ethical considerations

This study was approved by the Medical Ethics Committee, Ministry of Health, and Expert Committee at the School of Public Health, Mongolian National University of Medical Sciences. A written informed consent form was obtained from each participant before the interview and physical measurements.

### Data collection

Before data collection, all field members completed a training program on conducting the interview and certified nurses performed the blood draw, storing, and transporting for laboratory testing. A pilot study was conducted on five randomly selected PHCCs in November 2017. The data collection process was conducted between December 2017 and January

2018. In total, 4515 participants were included in the sample. The entire data collection procedure was conducted on an electronic tablet (Fire HD 8, Amazon, Seattle, WA, USA). To avoid data loss, an android application with an offline mode, QuickTapSurvey (TabbleDabble Inc., Toronto, Ontario, Canada), was used.

Interviews were conducted using a Mongolian version of the WHO STEPS Instrument for Noncommunicable Disease Risk Factor Surveillance [11]. To ensure the adequacy of the Mongolian translation of questionnaires, the Mongolian language versions were separately back-translated by two independent translators and reviewed.

## Measurements and definitions

Blood samples were collected from participants after an overnight fast for at least 10 hours. Fasting plasma glucose levels were measured by the biochemical method, and readings were interpreted based on WHO/IDF diagnosis criteria. Impaired fasting glycaemia (IFG), or pre-diabetes, was defined as fasting blood glucose (FBG) of 6.1–6.9 mmol/l. Diabetes was defined as (i) self-reported intake of sugar-lowering medication during the last two weeks and/or (ii) FBG equal to or above 7.0 mmol/l [12]. Socio-economical, behavioral, and metabolic risk factors were considered as predictor variables in diabetes prevalence. As the calculation of the diabetes prevalence heavily relies on the biochemical test results, we analyzed the data of 2169 people whose blood was drawn for laboratory analysis.

Body height and weight were measured according to a standard protocol, and body mass index (BMI) is calculated as weight in kilograms divided by height in meters squared. Overweight was defined as a person with a BMI between 25.0 to 29.9, and obese as a person with a BMI of 30 and higher.

Blood pressure was measured using the accuracy-validated BP A6 BTs (Microlife Corporation), digital automatic blood pressure monitors (Microlife Corporation, Taipei, Taiwan). A blood pressure reading of 130/80 mm Hg and over was considered as hypertensive according to the Mongolian Hypertension guideline that was updated in 2018 in line with AHA 2017 updated guideline.

To understand diabetes care, all diabetic participants were classified as either unaware, diagnosed but untreated, treated but uncontrolled, or controlled at the time of the study. Diabetic participants were classified as unaware if FBG was above the diabetes threshold (≥7.0 mmol/l) and they reported never having been diagnosed with diabetes. Participants were classified as untreated if questionnaire answers showed that they were aware of a diagnosis of diabetes but had not taken prescribed sugar-lowering medication in the last 14 days. Participants were classified as treated but uncontrolled if questionnaire answers showed that they had taken prescribed diabetes medication in the last 14 days but FBG ≥ 7.0 mmol/l. Participants were classified as controlled if questionnaire answers showed that they had taken prescribed antidiabetics in the last 14 days and FBG was lower than the diabetes threshold (≤7.0 mmol/l).

We classified the age of the participants into 6 groups (20–29, 30–39, 40–49, 50–59, 60–69. 70+) and age-and sex-specific prevalence of diabetes and prediabetes (IFG) was calculated. The overall prevalence of diabetes and prediabetes was estimated by standardizing the data according to the sex and age groups. The prevalence of fasting blood glucose levels was standardized using the WHO standard population structure to account for age and sex distribution. The levels of awareness of diagnosis, treatment, and control were expressed as percentages. Multivariate logistic regression investigated to what degree socio-economic, behavioral, anthropometric, and biochemical variables predicted diabetes prevalence. Statistical significance was defined as a P-value < 0.05. All the statistical analyses were conducted with R version 4.1.3 (One Push-Up) software.

## Results

### Distributions of fasting plasma glucose

Table 1 summarizes the sociodemographic characteristics of the sample. We analyzed the data from 2169 participants, comprising 1199 females (55.3%) and 970 males (44.7%). The mean age was 41.1*±14.1 years, with females averaging 41.9±14.1 and males 40.2±14.1 years.

**Table 1. Basic demographic characteristics of study participants.**

| Demographic factors | Female | Male | Overall |
|---|---|---|---|
| | 1199 (55.3%) | 970 (44.7%) | 2169 (100.0%) |
| Age Group (N = 2169) | | | |
| 20-29 | 294 (24.5%) | 271 (27.9%) | 565 (26.0%) |
| 30-39 | 293 (24.4%) | 260 (26.8%) | 553 (25.5%) |
| 40-49 | 244 (20.4%) | 180 (18.6%) | 424 (19.5%) |
| 50-59 | 207 (17.3%) | 156 (16.1%) | 363 (16.7%) |
| 60-69 | 126 (10.5%) | 73 (7.5%) | 199 (9.2%) |
| 70+ | 35 (2.9%) | 30 (3.1%) | 65 (3.0%) |
| Education (N = 2169) | | | |
| Higher Education | 620 (51.7%) | 422 (43.5%) | 1042 (48.0%) |
| No Higher Education | 579 (48.3%) | 548 (56.5%) | 1127 (52.0%) |
| Marital status (N = 2161) | | | |
| Married | 988 (82.4%) | 752 (77.5%) | 1740 (80.2%) |
| Not Married | 206 (17.2%) | 215 (22.2%) | 421 (19.4%) |
| Employment (N = 2148) | | | |
| Employed | 619 (51.6%) | 673 (69.4%) | 1292 (59.6%) |
| Unemployed | 562 (46.9%) | 294 (30.3%) | 856 (39.5%) |
| Income (N = 1914) | | | |
| <170,000 | 69 (5.8%) | 36 (3.7%) | 105 (4.8%) |
| 170,000 - 370,000 | 160 (13.3%) | 100 (10.3%) | 260 (12.0%) |
| 370,000 - 570,000 | 253 (21.1%) | 168 (17.3%) | 421 (19.4%) |
| 570,000 - 770,000 | 200 (16.7%) | 145 (14.9%) | 345 (15.9%) |
| 770,000 - 970,000 | 159 (13.3%) | 131 (13.5%) | 290 (13.4%) |
| 970,000< | 225 (18.8%) | 268 (27.6%) | 493 (22.7%) |

Note: The table shows the number and percentage of female and male participants across different demographic categories. Percentage are calculated within each gender group. Total sample size is 2169 unless otherwise indicated.

The overall weighted mean fasting plasma glucose was 5.33 ± 1.89 mmol/l. Mean plasma glucose was higher among males than among females: 5.45 mmol/l for men and 5.21 mmol/l for women. Plasma glucose increased with age: the age group with the lowest mean plasma glucose was 20–29 (4.80 mmol/l), while the age group with the highest mean plasma glucose was 50–59 (6.03 mmol/l). Mean plasma glucose among the 60–69 and 70 + age groups was lower than in the 50–59 group but was still higher than in the younger age groups (20–29, 30–39, 40–49). For females, the age group with the highest mean plasma glucose was 50–59 (6.11 mmol/l), while for males, the mean plasma glucose was highest among those older than 70 (6.38 mmol/l) (Table 2). Mean glucose calculated among non-diabetic, "relatively healthy" cohorts (<7.0 mmol/l) was 4.85 ± 0.6 mmol/l (4.78 ± 0.6 mmol/l among females, 4.93 ± 0.58 mmol/l for males).

### Prevalence of IFG and diabetes and risk factors associated with diabetes mellitus

A total of 175 participants had fasting blood glucose elevated between 6.1–6.9 mmol/l; thus, the prevalence of IFG in Ulaanbaatar was 8.1%. The prevalence of IFG among females was 6.6% (n = 79), while the prevalence of IFG among males was 9.9% (n = 96).

A total of 128 participants had fasting blood glucose higher than 7.0 mmol/l, with an additional 8 people reported having taken sugar-lowering medicine in the past 14 days. Thus, the overall prevalence of diabetes mellitus in Ulaanbaatar was 6.27% (n = 136) (Fig 1).

**Table 2. Distribution of mean fasting blood glucose, prevalence of pre-diabetes (IFG) and diabetes mellitus by age, sex category.**

| Sex | Age Group | Fasting blood glucose | | Glycemic status, n (%) | | |
|---|---|---|---|---|---|---|
| | | Mean (SD) | 95% CI | Diabetes (≥7.0 mmol/l) | IFG (6.1–6.9 mmol/l) | Healthy (<6.1 mmol/l) |
| Female | 20-29 | 4.64 (0.97) | 4.53 - 4.75 | 2 (3.4%) | 6 (7.6%) | 286 (26.9%) |
| | 30-39 | 4.92 (1.10) | 4.79 - 5.05 | 7 (12.1%) | 17 (21.5%) | 269 (25.3%) |
| | 40-49 | 5.2 (1.87) | 4.96 - 5.43 | 6 (10.3%) | 17 (21.5%) | 221 (20.8%) |
| | 50-59 | 6.11 (3.08) | 5.68 - 6.53 | 29 (50.0%) | 19 (24.1%) | 159 (15.0%) |
| | 60-69 | 5.8 (1.97) | 5.45 - 6.14 | 13 (22.4%) | 18 (22.8%) | 95 (8.9%) |
| | 70+ | 5.2 (0.71) | 4.96 - 5.45 | 1 (1.7%) | 2 (2.5%) | 32 (3.0%) |
| **Overall (N=1199)** | | **5.21 (1.89)** | **5.10 - 5.32** | **58 (4.8%)** | **79 (6.6%)** | **1062 (88.6%)** |
| Male | 20-29 | 4.97 (0.87) | 4.87 - 5.07 | 5 (7.1%) | 16 (16.7%) | 250 (31.1%) |
| | 30-39 | 5.33 (1.70) | 5.13 - 5.54 | 19 (27.1%) | 28 (29.2%) | 213 (26.5%) |
| | 40-49 | 5.45 (1.44) | 5.24 - 5.66 | 9 (12.9%) | 15 (15.6%) | 156 (19.4%) |
| | 50-59 | 5.94 (2.70) | 5.51 - 6.36 | 19 (27.1%) | 20 (20.8%) | 117 (14.6%) |
| | 60-69 | 6.27 (2.56) | 5.68 - 6.87 | 12 (17.1%) | 14 (14.6%) | 47 (5.8%) |
| | 70+ | 6.38 (3.39) | 5.11 - 7.64 | 6 (8.6%) | 3 (3.1%) | 21 (2.6%) |
| **Overall (N=970)** | | **5.45 (1.89)** | **5.33 - 5.57** | **70 (7.2%)** | **96 (9.9%)** | **804 (82.9%)** |
| Both sexes | 20-29 | 4.8 (0.94) | 4.72–4.88 | 7 (5.5%) | 22 (12.6%) | 536 (28.7%) |
| | 30-39 | 5.11 (1.43) | 5.00–5.23 | 26 (20.3%) | 45 (25.7%) | 482 (25.8%) |
| | 40-49 | 5.3 (1.7) | 5.14–5.47 | 15 (11.7%) | 32 (18.3%) | 377 (20.2%) |
| | 50-59 | 6.03 (2.92) | 5.73–6.34 | 48 (37.5%) | 39 (22.3%) | 276 (14.8%) |
| | 60-69 | 5.9 (2.21) | 5.66–6.28 | 25 (19.5%) | 32 (18.3%) | 142 (7.6%) |
| | 70+ | 5.74 (2.41) | 5.14–6.34 | 7 (5.5%) | 5 (2.9%) | 53 (2.8%) |
| **Total (N=2169)** | | **5.32 (1.89)** | **5.24 - 5.40** | **128 (5.9%)** | **175 (8.1%)** | **1866 (86%)** |

Note: Fasting blood glucose levels are expressed in millimoles per liter (mmol/L). Diabetes mellitus is defined as fasting blood glucose ≥7.0 mmol/l. Pre-diabetes (Impaired Fasting Glucose, IFG) is defined as fasting blood glucose between 6.1 and 6.9 mmol/l. Healthy glucose status is defined as fasting blood glucose <6.1 mmol/l. Mean values are presented with standard deviation (SD) and 95% confidence intervals (CI). Numbers (n) and percentages (%) are shown for each glycemic status within age and sex categories.

The prevalence of diabetes mellitus among females was 5.3% (n=63), while the prevalence of diabetes mellitus among males The older age groups (50–59, 60–69, 70+) had a higher prevalence of diabetes compared to the younger age groups (20–29, 30–39, 40–49). The age groups of 60–69 had the highest diabetes prevalence of 14.6%, while young people in their twenties had the lowest of 1.2%. However, the patterns of diabetes prevalence across the age groups among males and females do not coincide. The peak prevalence among females is attributed to the 50–59 age group (14.5%), while males aged above 70 years have the highest prevalence (20%) among male age groups. Males were observed to consistently have a higher prevalence than females across all age groups except the 50–59 age group (Fig 2).

The association between diabetes mellitus and its demographic and socioeconomic risk factors is summarized in Table 3. Overall, the prevalence of diabetes was shown to be significantly higher among males than among females (OR = 1.63, P=0.007). The diabetes prevalence also increased with age. Compared to the 20–29 age group, the 60–69 age group had the highest, statistically significant, risk of developing diabetes (OR = 14.51, P<0.001), while the 40–49 age group possessed the lowest risk, which was still statistically significant (OR=3.22, P=0.011). Marriage and unemployment were observed to be associated with a higher prevalence of diabetes (OR=3.74, *P<.001* and OR=1.94, *P<.001*). The lack of a high educational degree seems to coincide with the increased level of blood glucose; however, the association was not proven to be statistically significant. Income level was observed to be a lackluster predictor of diabetes mellitus (Table 3).

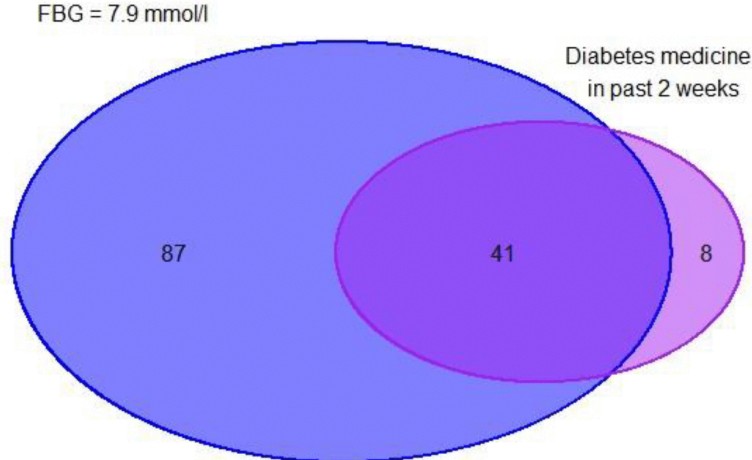

FBG = 7.9 mmol/l

Diabetes medicine in past 2 weeks

87    41    8

**Fig 1. Prevalence of diabetes mellitus among study participants.** Note: Blue circle represents participants with fasting blood glucose levels ≥7.0 mmol/l. The purple circle represents participants who have taken diabetes medicine in the past 2 weeks. Numbers within the circles indicate the count of participants in each category.

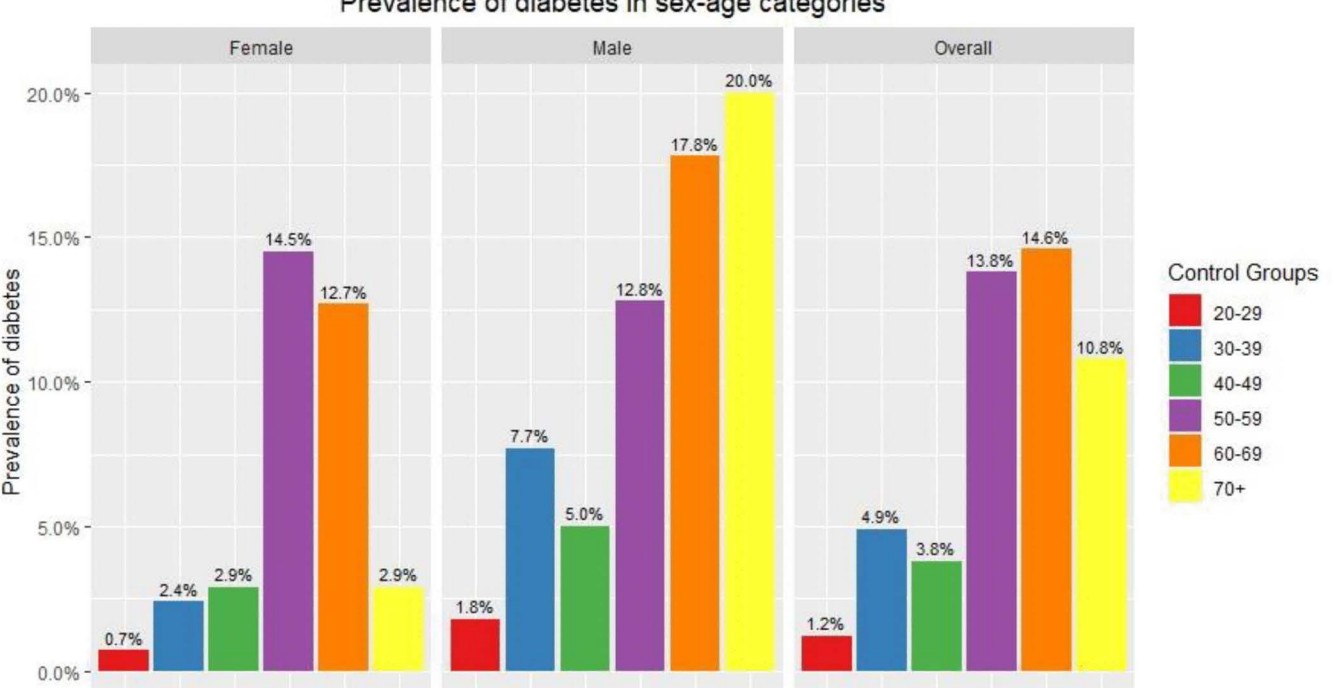

**Fig 2. Prevalence of diabetes in sex-age categories.** Note: This bar chart displays the percentage (%) of diabetes cases among different sex and age groups. The chart is divided into three sections: Female, Male, and Overall (both sexes combined). The axis shows the prevalence of diabetes as a percentage within each group.

**Table 3. Socio-economical risk factors associated with prevalence of diabetes mellitus.**

| Variables | Diabetic | % of diabetic participants | Prevalence in the category | adjusted OR | P-value |
|---|---|---|---|---|---|
| **Sex** | | | | | |
| Female | 63 | 46.3% | 5.3% | **1** | |
| Male | 73 | 53.7% | 7.5% | **1.63** | **0.007** |
| **Age (years)** | | | | | |
| 20-29 | 7 | 5.1% | 1.2% | **1** | |
| 30-39 | 27 | 19.9% | 4.9% | **4.12** | **<0.001** |
| 40-49 | 16 | 11.8% | 3.8% | **3.22** | **0.011** |
| 50-59 | 50 | 36.8% | 13.8% | **13.15** | **<0.001** |
| 60-69 | 29 | 21.3% | 14.6% | **14.51** | **<0.001** |
| 70+ | 7 | 5.1% | 10.8% | **9.76** | **<0.001** |
| **Education** | | | | | |
| Higher Education | 61 | 44.9% | 5.9% | 1 | |
| No Higher Education | 75 | 55.1% | 6.7% | 1.05 | 0.79 |
| **Marital status** | | | | | |
| Not Married | 8 | 5.9% | 1.9% | **1** | |
| Married | 128 | 94.1% | 7.4% | **3.74** | **<0.001** |
| **Employment** | | | | | |
| Employed | 60 | 44.1% | 4.6% | **1** | |
| Unemployed | 76 | 55.9% | 8.9% | **1.94** | **<0.001** |
| **Income** | | | | | |
| <170,000 | 7 | 5.1% | 6.7% | 1 | |
| 170,000 - 370,000 | 22 | 16.2% | 8.5% | 1.3 | 0.566 |
| 370,000 - 570,000 | 29 | 21.3% | 6.9% | 1.09 | 0.843 |
| 570,000 - 770,000 | 27 | 19.9% | 7.8% | 1.36 | 0.49 |
| 770,000 - 970,000 | 13 | 9.6% | 4.5% | 0.79 | 0.632 |
| 970,000< | 27 | 19.9% | 5.5% | 1.03 | 0.948 |
| **Overall** | **136** | **100.0%** | **6.3%** | | |

The association between diabetes mellitus and its behavioral and metabolic risk factors is summarized in Table 4. Out of metabolic risk factors, raised triglyceride (TG) level and decreased level of high-density lipoproteins (HDL) were observed to be significantly associated with diabetes (OR=3.24, P<.001 and OR=3.02, P=0.003). Being overweight or obese significantly raised the risk of developing diabetes mellitus (OR=2.8, P<.001 and OR=4.77, P<.001). Hypertension, or increased arterial pressure, was a strong predictor of diabetes (OR=3.75, P=0.002) for both sexes and across all age groups. Behavioral factors, such as smoking and drinking habits or frequency of fruit and vegetable consumption, were not observed to be significantly associated with the disease (Table 4). The separate risk factor analyses for females and males show that, although BMI was a significant predictor of diabetes for both sexes, high BMI had a stronger association with diabetes among females than among males. The odds of developing diabetes for overweight women were 5.77 (P<.001), while the odds for women suffering from obesity were 9.37 (P<0.001). The odds of developing diabetes for men suffering from obesity were 3.05 (P<.001) while being overweight was not significantly associated with diabetes for men. Among males, a decreased HDL level was the significant metabolic risk factor (OR=4.72, P=0.002); TG level was not observed to be significantly associated with diabetes in this group. Among females, an increased TG level was the significant metabolic factor (OR=6.36, P<0.001); HDL level was not observed to be significantly associated with diabetes in this group.

**Table 4. Behavioral and metabolic risk factors associated with prevalence of diabetes mellitus.**

| Variables | Diabetic (N = 128) | % of diabetic participants | adjusted OR | P-value |
|---|---|---|---|---|
| **Smoking** | | | | |
| Non-smoker | 93 | 68.4% | 1 | |
| Smoker | 43 | 31.6% | 0.98 | 0.952 |
| **Alcohol drinking** | | | | |
| Non-drinker | 21 | 15.4% | 1 | |
| Drinker | 46 | 33.8% | 0.85 | 0.537 |
| **Fruits, vegetable consumption** | | | | |
| ≥ 5 Units/day | 21 | 15.4% | 1 | |
| < 5 Units/day | 115 | 84.6% | 0.81 | 0.52 |
| **Biochemical indicators** | | | | |
| **Total cholesterol** | | | | |
| < 5 mmol/l | 47 | 34.6% | 1 | |
| ≥ 5 mmol/l | 89 | 65.4% | 1.79 | 0.124 |
| **Triglyceride** | | | | |
| < 1.7 mmol/l | 52 | 38.2% | **1** | |
| ≥ 1.7 mmol/l | 83 | 61.0% | **3.24** | **<0.001** |
| **HDL** | | | | |
| > 1.03 mmol/l (male) > 1.29 mmol/l (female) | 102 | 75.0% | **1** | |
| < 1.03 mmol/l (male) < 1.29 mmol/l (female) | 34 | 25.0% | **3.02** | **0.003** |
| **LDL** | | | | |
| < 3.3 mmol/l | 34 | 25.0% | 1 | |
| 3.30–4.1 mmol/l | 9 | 6.6% | 0.48 | 0.102 |
| ≥ 4.1 mmol/l | 6 | 4.4% | 0.76 | 0.597 |
| **Hypertension** | | | | |
| Non-hypertensive | 6 | 4.4% | **1** | |
| Hypertensive | 130 | 95.6% | **3.75** | **0.002** |
| **BMI** | | | | |
| Normal | 26 | 19.1% | **1** | |
| Overweight | 57 | 41.9% | **2.8** | **<0.001** |
| Obese | 53 | 39.0% | **4.77** | **<0.001** |

Note:This table presents the distribution of diabetes cases among study participants by various socio-economic factors.

## Awareness, treatment, and control of diabetes mellitus

Among all diabetic patients (n = 136), 48.5% (n = 66) were unaware they had diabetes, 15.4% (n = 21) were aware of diagnosis but untreated, 30.2% (n = 41) were treated but uncontrolled, and only 5.9% (n = 8) were controlled (Table 5). The control rate among the diabetics on sugar-lowering treatment was 16.3% (n = 8).

The unaware group was relatively younger compared to those aware, treated, and controlled. The median age of the diabetics who were unaware of their diagnosis was 49. The median ages of those who were aware but untreated, treated but uncontrolled, and controlled were 56, 55, and 60, respectively (Fig 3).

Among the male diabetics, the majority (63.0%) were unaware of their diagnosis. 9.6% of male participants with diabetes mellitus were aware of their diagnoses but never received treatment. 23.3% opted for treatment but never had

**Table 5. Awareness, treatment, and control of diabetes mellitus.**

| DIABETIC (N = 136) | |
|---|---|
| **Awareness** | |
| *Aware* | *Unaware* |
| 70 (51.47%) | 66 (48.53%) |
| **Treatment** | |
| *Treated* | *Untreated* |
| 49 (36.03%) | 21 (15.44%) |
| **Control** | |
| *Controlled* | *Uncontrolled* |
| 8 (5.88%) | 41 (30.15%) |

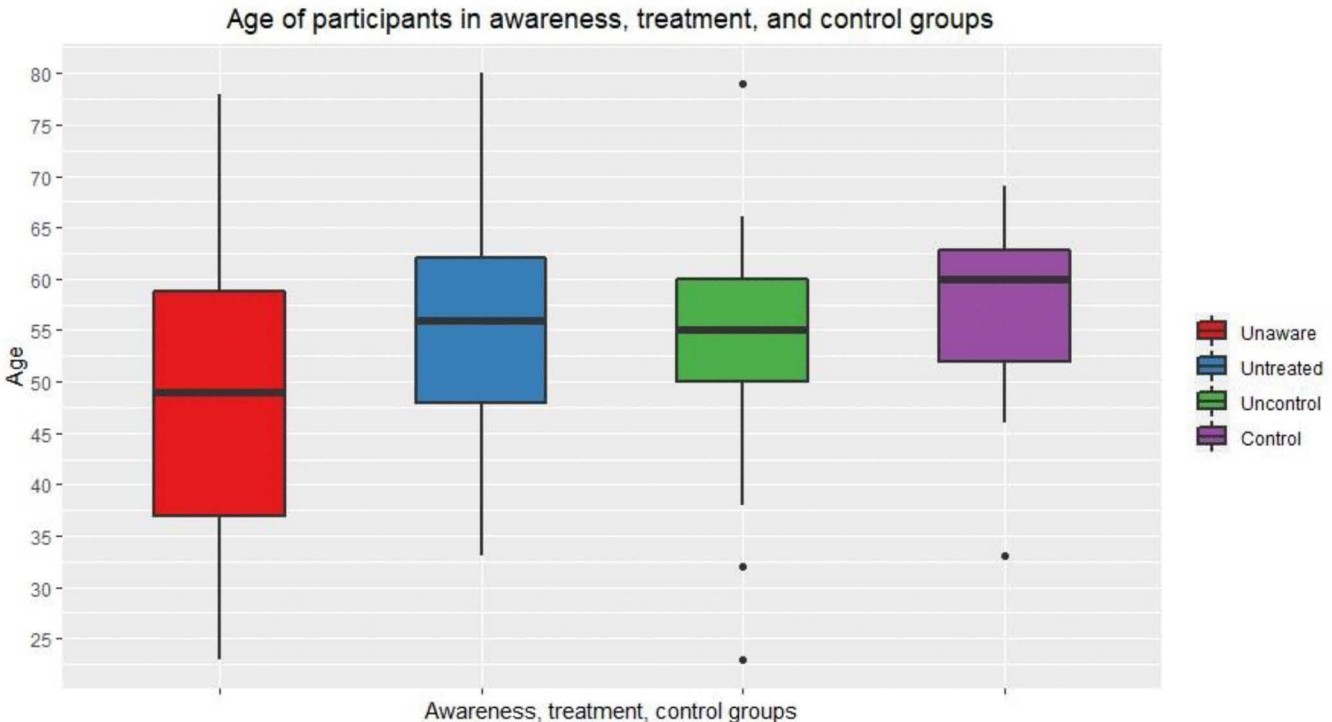

**Fig 3. Age of participants in awareness, treatment, and control groups.**

their treatment controlled, and only 4.1% received controlled treatment. As for the female diabetics, 31.7% were unaware of their diagnosis, while 22.2% were previously diagnosed but untreated. In female diabetics in our study, the highest percentage (38.1%) comprised the treated but uncontrolled group, while the remaining 7.9% accounted for the control rate (Fig 4).

The analysis of the relationship between awareness of diagnosis and sociodemographic factors showed that the unaware-ness rate was significantly higher among men than among women (63.0% vs 31.7%, OR=3.66, P<0.001). The awareness level was presumably higher among older age groups: compared to the 60–69 age group, the 20–29 and 30–39 age groups had significantly higher rate of unawareness (85.7% vs 27.6%, OR=13.76, P<0.05; 74.1% vs 27.6%, OR=6.14, P<0.05).

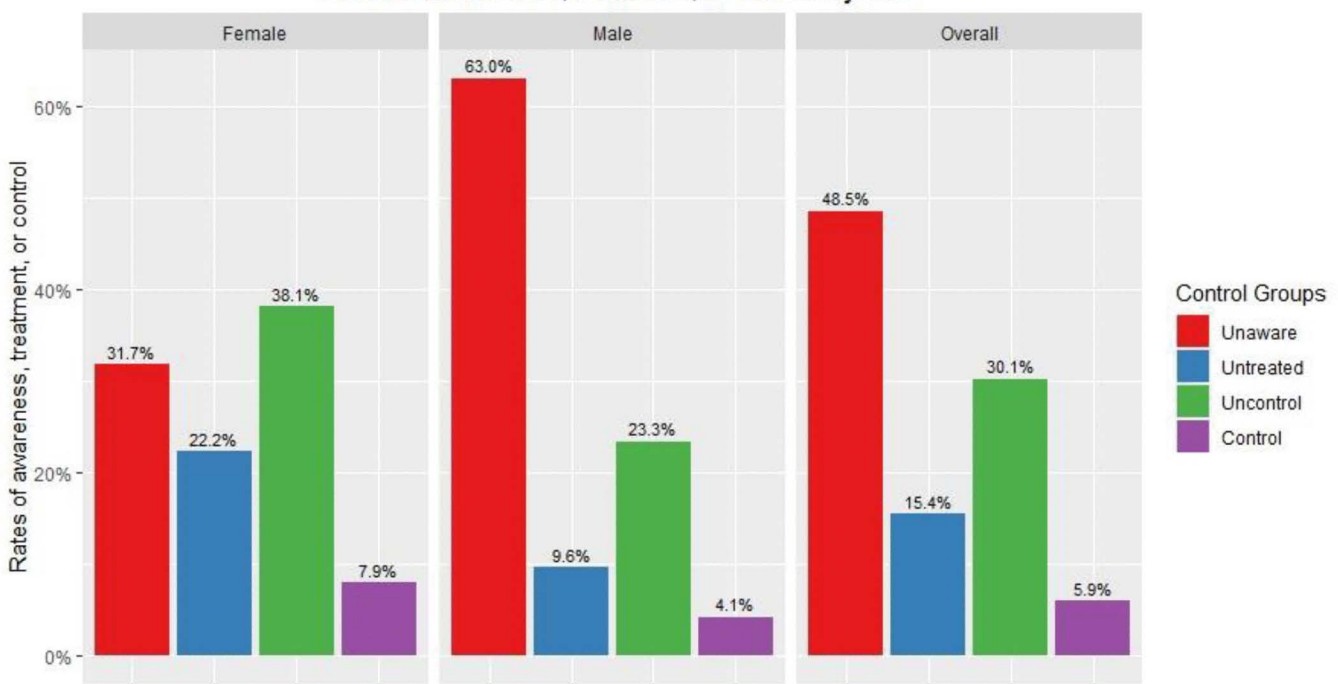

**Fig 4. Awareness, treatment, and control by sex.**

The unawareness level rose again among the 70+ age group (71.4%); however, it was not observed to be statistically different from the unawareness rate calculated for the 30–39 age groups. There was also no statistically significant difference between the two youngest age groups. Participants with no high educational degree were more likely to be unaware of their diagnosis compared to people who carried high educational degrees (61.3% vs 32.8%, OR=3.25, P<0.001). There were also significantly more people who were unaware that they had diabetes among the employed group of participants compared to the unemployed group (60.0% vs 39.5%, OR=2.3, P<0.05). The unawareness rate decreased with the increase in income. In addition, the unawareness rate among married participants was higher than among single people. However, income and marital status were not observed to be statistically significant factors of awareness of the diagnosis (Table 6).

The analysis of the relationship between sociodemographic factors and the treatment rate among the participants who had been aware of their diagnosis before the study showed no statistically significant association between any of the factors and the choice of receiving treatment (Table 6). The percentage of diagnosed but untreated people was observed to be higher among females compared to males, and single people compared to the married (22.2% vs 9.6%; 25.0% vs 14.8%). The rate of opting out of treatment with the knowledge of diagnosis was also higher among the unemployed compared to those employed (19.7% vs 10.0%). No participant in the 20–29 age group was detected to be diagnosed but untreated: the only participant in their twenties who was reported to have been aware of their diagnosis was receiving treatment. The treatment rates in other age groups were observed to be within the close range. There was no participant from the lowest income group that belonged to the diagnosed but untreated group: the two participants from the lowest income group that were aware of their diagnosis were also receiving treatment. The treatment rates in the income groups were mostly within the close range; however, the percentage of diagnosed but untreated people rose distinguishably in the highest income group.

**Table 6. Awareness, treatment, and control of diabetes by sociodemographic categories.**

| Socio-Economic categories | Unaware | | | Diagnosed, untreated | | | Treated, uncontrolled | | |
|---|---|---|---|---|---|---|---|---|---|
| | n | % in the category (among diabetics) | OR | n | % in the category (among diabetics) | OR | n | % in the category (among diabetics) | OR |
| **Sex** | | | | | | | | | |
| Female | 20 | 31.7% | 1 | 14 | 22.2% | 1.38 | 24 | 38.1% | 1 |
| Male | 46 | 63.0% | **3.66**** | 7 | 9.6% | 1 | 17 | 23.3% | 1.18 |
| **Age (years)** | | | | | | | | | |
| 20-29 | 6 | 85.7% | **13.76*** | 0 | 0.0% | NA | 1 | 14.3% | NA |
| 30-39 | 20 | 74.1% | **6.14*** | 3 | 11.1% | 3.02 | 3 | 11.1% | 1 |
| 40-49 | 7 | 43.8% | 1.89 | 3 | 18.8% | 1.68 | 5 | 31.3% | 2.43 |
| 50-59 | 20 | 40.0% | 1.91 | 9 | 18.0% | 1.29 | 19 | 38.0% | 4.75 |
| 60-69 | 8 | 27.6% | 1 | 5 | 17.2% | 1 | 12 | 41.4% | 1.38 |
| 70+ | 5 | 71.4% | 4.63 | 1 | 14.3% | 4.84 | 1 | 14.3% | NA |
| **Marital status** | | | | | | | | | |
| Not Married | 3 | 37.5% | 1 | 2 | 25.0% | 1.61 | 2 | 25.0% | 1 |
| Married | 63 | 49.2% | 1.62 | 19 | 14.8% | 1 | 39 | 30.5% | 2.79 |
| **Education** | | | | | | | | | |
| Higher Education | 20 | 32.8% | 1 | 11 | 18.0% | 1 | 23 | 37.7% | 1 |
| No Higher Education | 46 | 61.3% | **3.25**** | 10 | 13.3% | 1.44 | 18 | 24.0% | 5.48 |
| **Income** | | | | | | | | | |
| <170,000 | 5 | 71.4% | 1 | 0 | 0.0% | NA | 2 | 28.6% | 1 |
| 170,000 - 370,000 | 13 | 59.1% | 0.58 | 3 | 13.6% | 1 | 6 | 27.3% | NS |
| 370,000 - 570,000 | 17 | 58.6% | 0.57 | 4 | 13.8% | 1 | 7 | 24.1% | NS |
| 570,000 - 770,000 | 11 | 40.7% | 0.28 | 4 | 14.8% | 0.67 | 9 | 33.3% | NS |
| 770,000 - 970,000 | 5 | 38.5% | 0.25 | 2 | 15.4% | 0.67 | 3 | 23.1% | NS |
| >970,000 | 11 | 40.7% | 0.28 | 7 | 25.9% | 1.56 | 9 | 33.3% | NS |
| **Employment** | | | | | | | | | |
| Employed | 36 | 54.5% | **2.3*** | 6 | 10.0% | 1 | 14 | 23.3% | 1 |
| Unemployed | 30 | 45.5% | 1 | 15 | 19.7% | 1.45 | 27 | 35.5% | 1.93 |
| **Overall** | **66** | **48.5%** | | **21** | **15.4%** | | **41** | **30.1%** | |

*P-value < .05.

**P-value < .001.

NS-non-significant.

NA-not analyzed due to few observations.

Treatment was not observed to result in any statistically significant change in lifestyle, anthropometric, and biochemical indicators among the previously diagnosed participants. A larger percentage of treated people had a safe level of total cholesterol and low-density lipoproteins compared to those untreated (34.7% vs 19.0%; 30.6% vs 9.5%). Every participant in the diagnosed but untreated group had hypertension. The only non-hypertensive participant in the diagnosed group had been receiving treatment. All the diagnosed but untreated participants consume alcohol in certain amounts. Out of treated participants, six had reported themselves as abstainers from alcoholic drinks (Table 7).

There is no statistically significant association between any demographic factor and the control/uncontrol rate (Table 6). The percentage of treated but uncontrolled people was higher among female diabetics compared to their male counterparts (38.1% vs 23.3%). The uncontrol rate among diabetics was also higher for the 40–49, 50–59, 60–69 age groups compared

**Table 7. Behavioral and metabolic indicators within treatment and control groups.**

| | Diagnosed | | Treated | |
| --- | --- | --- | --- | --- |
| | Diagnosed and treated (n = 49) | Diagnosed but untreated (n = 21) | Treated but uncontrolled (n = 41) | Treated and controlled (n = 8 |
| **Smoking** | | | | |
| Yes | 11 (22.4%) | 5 (23.8%) | 9 (22.0%) | 2 (25.0%) |
| No | 38 (77.6%) | 16 (76.2%) | 32 (78.0%) | 6 (75.0%) |
| **Alcohol** | | | | |
| Yes | 43 (87.8%) | 21 (100%) | 36 (87.8%) | 7 (87.5%) |
| No | 6 (12.2%) | 0 (0%) | 5 (12.2%) | 1 (12.5%) |
| **Total cholesterol** | | | | |
| ≥5 mmol/l | 32 (65.3%) | 17 (81.0%) | 29 (70.7%) | 3 (37.5%) |
| <5 mmol/l | 17 (34.7%) | 4 (19.0%) | 12 (29.3%) | 5 (62.5%) |
| **Triglyceride** | | | | |
| ≥1.7 mmol/l | 33 (67.3%) | 13 (61.9%) | 29 (70.7%) | 4 (50.0%) |
| <1.7 mmol/l | 16 (32.7%) | 8 (38.1%) | 12 (29.3%) | 4 (50.0%) |
| **HDL** | | | | |
| <1.03mmol/l (male) <1.29 mmol/l (female) | 16 (32.7%) | 7 (33.3%) | 15 (36.6%) | 1 (12.5%) |
| >1.03 mmol/l (male) >1.29 mmol/l (female) | 33 (67.3%) | 14 (66.7%) | 26 (63.4%) | 7 (87.5%) |
| **LDL** | | | | |
| ≥3.3 mmol/l | 34 (69.4%) | 19 (90.5%) | 30 (73.2%) | 4 (50.0%) |
| <3.3 mmol/l | 15 (30.6%) | 2 (9.5%) | 11 (26.8%) | 4 (50.0%) |
| **Hypertension** | | | | |
| Hypertensive | 48 (98.0%) | 21 (100%) | 40 (97.6%) | 8 (100%) |
| Healthy | 1 (2.0%) | 0 (0%) | 1 (2.4%) | 0 (0%) |
| **BMI** | | | | |
| Overweight | 43 (87.8%) | 17 (81.0%) | 35 (85.4%) | 8 (100%) |
| Normal | 6 (12.2%) | 4 (19.0%) | 6 (14.6%) | 0 (0%) |

to the younger age groups. Married diabetics had a higher uncontrol rate compared to those not married (30.5% vs 25.0%). The uncontrol rate was higher among those with a higher educational degree than among those with no high educational degree (37.7% vs 24.0%). The uncontrol rates in the income groups were observed to be within the close margin (23–33%); however, the highest income group had the highest rate (33.3%). The percentage of diabetics who received treatment but remained uncontrolled was higher among the unemployed compared to those with a certain occupation (35.5% vs 23.3%).

Control was not observed to result in any statistically significant change in lifestyle, anthropometric, and biochemical indicators among the diagnosed and treated participants. There was a higher percentage of people with healthy levels of total cholesterol, triglyceride, HDL, and (low-density lipoprotein) LDL among the controlled diabetics compared to the uncontrolled. There was no non-hypertensive participant in the controlled group. The only participant who was not suffering from elevated blood pressure did not get their diabetic treatment controlled. There was also no participant with a normal BMI in the control group. All six treated participants with a normal BMI had not gotten their diabetes controlled (Table 7).

## Discussion

This study found that the average prevalence of type II diabetes in Ulaanbaatar is lower than the average prevalence of diabetes among middle income countries (6.27% vs 10.8%) [3]. Moreover, it is lower compared to the diabetes prevalence

rates recorded in other Asian countries: 9.2% in Bangladesh [13], 10.2% in Myanmar [14], 9.9% in Thailand [15], 12.5% in Kazakhstan [16], 7.9% in Uzbekistan [17], and 12.4% in neighboring China [18]. However, despite the relatively low prevalence of diabetes mellitus, the control rate of diabetes in Ulaanbaatar is far from optimal. It is worth that other studies in Mongolia have reported higher national prevalence estimates than observed in our study. For example, Dayan et all reported an age-standardized diabetes prevalence of approximately 10.0% among Mongolian adults aged 30 and over, using a nationally representative sample [19]. This is notably higher than our Ulaanbaatar estimate of 6.27%, suggesting that urban-rural or demographic differences may influence diabetes burden across the country. The awareness, treatment, and control rates for adults in Laos were at 58.1%, 40.3%, 10.9% [20], in Bangladesh at 41.2%, 36.9%, 14.2% [13], in China at 36.7%, 32.9%, 16.5% [18], while the corresponding rates in Ulaanbaatar were 51.5%, 36%, and 5.9% among all diabetics. The control rate among the treated diabetics was 16.3%.

Sex and gender were estimated to be good predictors of T2DM in Ulaanbaatar. Diabetes in Ulaanbaatar was more prevalent among men than among women, which matches a worldwide trend observed worldwide. A meta-analysis conducted by the NCD Risk Factor Collaboration on 751 population-based studies (4.4 million adults from 146 countries) reported that prevalence rates of diabetes mellitus type 2 more markedly increased in men (4% to 9%) than in women (5% to 8%) between 1980 and 2014, despite some disparities between certain geographical areas [21]. The latest Diabetes Atlas from International Diabetes Federation reports that the prevalence of diabetes is slightly lower among women than among men, and there were 17.7 million more men than women living with diabetes in 2021 [3].

The prevalence of diabetes in Ulaanbaatar increased with age. The age group of 60–69 had the highest diabetes prevalence among all participants, while those in their twenties were observed to have the lowest diabetes prevalence. Our findings reflect the description of 2021 global diabetes estimates reported in the latest IDF Diabetes Atlas. However, by these estimates, the peak prevalence was attributed to the 50–59 age group [3]. Recognizing age as one of the major risk factors for diabetes, the American Diabetes Association recommends screening for type 2 diabetes for all adults 45 years and older regardless of the presence of other risk factors [22].

Although the logistic regression analysis showed a positive association between marriage and diabetes, we believe the two variables were confounded by age. The older participants were significantly more likely to be married than the younger participants. While the ratio of married participants to those not married in the 20–29 age group was close to 1:1, this ratio became 9:1 among participants 40 years and older. Therefore, we believe the association between marriage and diabetes is tied to the association between older age and diabetes.

Employed participants were more likely to have diabetes compared to the unemployed. Nevertheless, we suspect that this observation may have been a case of a reverse effect. Diabetes is included in the list of diseases to be used to diagnose disability that was attached to the joint order of the Minister of Health and the Minister of Labor and Social Welfare dated December 9, 2021. Diabetics with a compromised regulation of blood glucose are estimated to have lost 60% of their ability to work and are qualified to receive social support for up until one year until they have their wellbeing restored. Diabetics with late-stage complications qualify for longer years of social support: those with microvascular complications can receive support for a year, while those with macrovascular complications qualify for two years. If by the end of the period the individual's health doesn't stabilize, they can apply for the subsidy repeatedly. Diabetics suffering from chronic organ failure resulting from macro and microvascular complications are considered to have lost 80% of their ability to work and can receive social support for an indefinite period [23]. Our data lack information on any complications experienced by those diagnosed with diabetes or any social support they might have been receiving. Nevertheless, with the positive association between awareness and unemployment, as well as the low control rate, we are supposing that a major percentage of diabetics are unemployed, as they might qualify for a disability aid due to a lack of blood glucose regulation and complications stemming from poor control.

Diabetes is often a comorbid disease that confounds with hypertension. The results of a large prospective cohort study by Gress et. al., including 12550 adults, showed that hypertensive people were 2.5 times more likely to develop type II

diabetes compared to normotensive people [24]. In Ulaanbaatar, diabetes was 3.75 times as likely among the hypertensive compared to their healthy counterparts. Diabetes, coupled with hypertension, significantly increases the risk of serious cardiovascular and renal diseases.

Other major risk factors for CVDs in people with diabetes include obesity and dyslipidemia. Compared to people with normal BMI, the overweight had a 2.8 times higher chance of having diabetes, while those suffering from obesity were 4.8 times more likely to develop diabetes. A review by Verma and Hussain explains in great detail the pathophysiology between obesity and diabetes. Obesity results in a long-term increase in plasma free fatty acids (FFA), which in turn leads to insulin deficiency and insulin resistance, the two key causes of diabetes [25].

TG and HDL levels were significantly associated with type 2 diabetes in Ulaanbaatar. Changes with these lipid levels, specifically higher concentrations of TG and lower concentrations of HDL, were reported to be linked with aforementioned insulin resistance according to a review authored by Krauss and Siri [26]. Categorically, an insufficient level of HDL seemed to have a stronger association with diabetes among males, while increased TG was linked with diabetes in females. Walden et.al. studied sex differences in the association between diabetes mellitus and lipoprotein and triglyceride concentrations. The study suggested that diabetes was more related to adverse levels of TG and lipoprotein concentrations among women than among men [27]. Verification and further investigation of the association between a decreased HDL level and diabetes among men in Ulaanbaatar is recommended.

Appropriate glycemic control is vital for delaying or preventing diabetes complications altogether; however, diabetes control in Ulaanbaatar remains far from optimal. The findings above suggest that the most significant barrier to increasing the diabetes control rate is the lack of knowledge of glycemic status among people, or, in other words, the high rate of undiagnosed diabetes. Almost half of the diabetic subjects were unaware they had diabetes prior to the study. This unawareness rate was significantly higher among men, young people aged between 20–39, people with no high education, and the employed. The hypothesis is that most people are diagnosed with diabetes after an occurrence of health complications. This could explain why, despite the unemployment of a significant proportion of diabetics, a majority of previously undiagnosed diabetics were employed.

The percentage of undiagnosed male diabetics among male diabetics is dramatically higher than the percentage of undiagnosed female diabetics among female diabetics. Men are generally less likely to utilize healthcare compared to women, resulting in men being at a higher risk of mortality and morbidity – a sex pattern reported in various studies worldwide. The fundamental reason behind the gender disparities in health status is that men are less likely to access primary and preventative healthcare [28]. However, the rate of primary healthcare utilization among men increases significantly after their hospitalization [29]. These findings suggest that men are more willing to seek medical care after experiencing severe complications.

People without a high educational degree were also more likely to have been unaware of their diagnosis prior to the study. Diabetes is a relatively unknown disease in Mongolia: the general population lacks extensive knowledge about the disease. The study conducted by Demaio et al. on KAP related to diabetes in Mongolia reports that access to counseling and health education on diabetes was lowest for the least educated households (5). We posit that the limited presence of diabetes-related education has contributed to the population's low awareness levels. Mongolians are likely to turn to medical care and thus receive diagnosis of diabetes only after experiencing certain discomfort.

Diabetes decreases the likelihood of employment and increases absenteeism and work limitations [30]. With the emergence of health complications, the ability to work declines while the economic cost of diabetes grows. Thus, a low diabetes early detection rate is detrimental to not only the physical well-being of people but also their social and financial security.

The performance of the healthcare system for diabetes management in Ulaanbaatar generally can be characterized by insufficiency of quality care at every stage of the cascade: screening, diagnosis, treatment, and control. This insufficiency is related to factors from both the demand and supply sides. These include the absence of

specific national guidelines for diabetes management at the primary care level, limiting standardization and consistency of care. In addition, there are limited resources for training and upskilling primary care providers in non-communicable disease management. Finally, the lack of an integrated health information system hinders patient tracking and coordination across levels of care, reducing the efficiency and continuity of diabetes management service.

Lack of public awareness and low patient engagement are the demand-side factors. Measures to raise the public's knowledge about diabetes are fundamental, especially when targeting young men and those with no high education. Moreover, every patient, upon the diagnosis, should receive diabetes self-management education that encourages diabetics to actively and constantly engage with the treatment and control process. However, patient engagement is often hurdled by drawbacks in the healthcare system. The major reasons behind such a low control rate for diabetes lay within the healthcare system. There are no distinctive algorithms for diabetes care and no set criteria used to evaluate the diabetic patient's health performance and possible complications. Another obstacle is the absence of a continuous monitoring system.

Common diabetes medications, such as metformin, glimepiride, and gliclazide, are included in the list of essential drugs subsidized by the Health Insurance Fund of Mongolia. They can be accessed through a primary physician monthly. This system can become a foundation for a monitoring system that will improve diabetes control. While obtaining the prescription of the subsidized medicine, the patient could also have their health performance evaluated and, in case of any deviation from control standards, be directed to the appropriate treatment or another specialist. To build this system, it is essential to set algorithms for diabetes care and performance criteria. Moreover, further obstacles and challenges in primary diabetes care shall be investigated. In addition, it is crucial to evaluate the accessibility of diabetes medication and self-monitoring devices within and outside the national health insurance system.

A major strength of this study is the use of the WHO STEPS methodology, unsuring standardized data collection and comparability with global datasets. The questionnaire used was validated and previously applied across multiple populations [11],supporting the reliability of the findings. However, several limitations should be explicitly acknowledged: Although the study was adequately powered for population-level estimates, the sample size may have limited the ability to detect statistically significant differences in smaller subgroups. One district of Ulaanbaatar (Bagakhangai) was exluded due to its isolated geographical location. While this exclusion was logistically justified, it may have introduced a degree of selection bias and limits the generalizability of findings to all districts. In addition, although the sampling was population-based, the potential for non-response bias remains-particularly among working-age adults and men, who may be underrepresented due to to availability during data collection. This study was cross-sectional, which limits the ability to determine causal relationship between risk factors and diabetes outcomes. Another limitation of our study is that we applied the fasting blood glucose (FBG) test rather than the hemoglobin A1C (HbA1c) test to detect diabetes. Compared to FBG, HbA1c is more reliable for the identification of impaired glucose tolerance and evaluation of the levels of control and diagnosis of diabetes. However, we found employing the HbA1c test difficult as the actual HbA1c threshold value differed between various studies. In addition, WHO/IDF were concerned that HbA1c test results could be biased, as they can be influenced by several factors including anemia, abnormalities of hemoglobin, pregnancy, and uremia, which were not included in our study questionnaire. We did not collect data on diabetes-related complications, durations of disease, medication adherence, or access to disability/social support programs. These unmeasured variables may influence observed associations, particularly between employment and diabetes.

Diabetes is one of the major contributors to CVD and, being interconnected with elevated blood pressure, contributes to the burden of hypertension. Diagnosis, treatment, and control of diabetes in Ulaanbaatar are at insufficient levels. More research should be conducted to investigate barriers to better diabetes care, especially on factors affiliated with the healthcare system in Mongolia.

 

## Acknowledgments

We gratefully acknowledge the efforts of those current and past Onom Foundation staff, who contributed to conceiving the approach, planning and implementing the field work, and carried out the initial analyses.

## Author contributions

**Conceptualization:** Andreas Bungert, Naranbaatar Dashdorj.

**Data curation:** Maral Myanganbayar, Andreas Bungert.

**Formal analysis:** Uurtsaikh Baatarsuren, Khulan Tuvdendarjaa.

**Methodology:** Uurtsaikh Baatarsuren, Khulan Tuvdendarjaa, Maral Myanganbayar.

**Project administration:** Khulan Tuvdendarjaa, Maral Myanganbayar.

**Resources:** Delgerbat Boldbaatar.

**Supervision:** Myagmartseren Dashtseren, Tsolmon Unurjargal, Naranbaatar Dashdorj, Naranjargal Dashdorj.

**Writing – original draft:** Orkhonselenge Davaadamdin, Bolor-Erdene Ganbat.

**Writing – review & editing:** Yanfang Su, Tsolmon Unurjargal, Johannes Boch.

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
