## [Decision Letter · Decision Letter 0]

7 Apr 2025

Dear Dr. Dashdorj,

Thank you for submitting your manuscript to PLOS ONE. After careful consideration, we feel that it has merit but does not fully meet PLOS ONE’s publication criteria as it currently stands. Therefore, we invite you to submit a revised version of the manuscript that addresses the points raised during the review process.

We look forward to receiving your revised manuscript.

Kind regards,

Patricia Khashayar

Academic Editor

PLOS ONE

Journal Requirements:

2. Please note that your Data Availability Statement is currently missing the repository name and the DOI/accession number of each dataset or a direct link to access each database. If your manuscript is accepted for publication, you will be asked to provide these details on a very short timeline. We therefore suggest that you provide this information now, though we will not hold up the peer review process if you are unable.

**Additional Editor Comments:**

The main concern was regarding the method section, so more focus should be directed towards that section

Reviewers' comments:

Reviewer's Responses to Questions

**Comments to the Author**

1. Is the manuscript technically sound, and do the data support the conclusions?

Reviewer #1: Yes

Reviewer #2: Partly

2. Has the statistical analysis been performed appropriately and rigorously?

Reviewer #1: Yes

Reviewer #2: I Don't Know

3. Have the authors made all data underlying the findings in their manuscript fully available?

Reviewer #1: Yes

Reviewer #2: Yes

4. Is the manuscript presented in an intelligible fashion and written in standard English?

Reviewer #1: Yes

Reviewer #2: Yes

Reviewer #1: Line 58, While IFG is a criterion for diagnosing, it's not synonymous with diabetes itself (which can also be diagnosed via HbA1c as well). It would be better to include the exact definition and cut-off points of diabetes from STEPS.

Line 62, the KAP survey data is from 2010. While it highlights a historical lack of awareness, it is dated. If more recent studies are available, please include and cite them as well. If not, briefly acknowledge the age of this study while stating it is the best available indicator of past awareness levels.

Line 96, “For the laboratory analysis, one in two of the selected participants aged were recruited” Please specify the exact age range subjected to this selection for lab tests and provide some explanation about how the one in two were selected. Was it random or based on other criteria?

Line 151, Specify if any standard population structure were used to standardize the prevalence (WHO standard population etc.)

Table3, “parity” section seems incorrect considering the subheadings area marital status.

Line 400, fix "the with the overall minimal presence"

Line 407, Could you briefly explain in the manuscript on why these supply-side gaps might exist in the Mongolian context. Is it lack of specific national guidelines for diabetes management in primary care? Limited resources for training primary care physicians? Issues with laboratory access/standardization for monitoring (relevant given the HbA1c limitation discussion)? Lack of integrated health information systems?

Reviewer #2: Good paper with enough information. However there are some points needed to modify. Introduction: the paragraphs are not related to each other. Please make them in better order with relation sentences at the end of the each paragraph. Method: it is not clear what do you want to do. Please explain more about the method section. Result: I think this section is not in good order. Please clarify the outcome measures and explain the results in appropriate way. Discussion: I did not see the strength and limitation of your study.

**Do you want your identity to be public for this peer review?** For information about this choice, including consent withdrawal, please see our Privacy Policy

Reviewer #1: No

Reviewer #2: **Yes: ** Laleh Abadi marand

---

## [Author Response · Author response to Decision Letter 1]

21 May 2025

Dear Editors,

Thank you for the opportunity to revise our manuscript and for the thoughtful and constructive comments provided by the reviewers. We appreciate the time and effort invested in evaluating our work. We have carefully considered all the comments and revised the manuscript accordingly. In rebuttal letter, we provide a detailed, point-by-point response to each comment.

---

## [Decision Letter · Decision Letter 1]

14 Aug 2025

Dear Dr. Dashdorj,

We look forward to receiving your revised manuscript.

Kind regards,

Patricia Khashayar

Academic Editor

PLOS ONE

Journal Requirements:

Reviewers' comments:

Reviewer's Responses to Questions

**Comments to the Author**

Reviewer #3: All comments have been addressed

2. Is the manuscript technically sound, and do the data support the conclusions?

Reviewer #3: Yes

3. Has the statistical analysis been performed appropriately and rigorously?

Reviewer #3: Yes

4. Have the authors made all data underlying the findings in their manuscript fully available?

Reviewer #3: Yes

5. Is the manuscript presented in an intelligible fashion and written in standard English?

Reviewer #3: Yes

Reviewer #3: Clarify the Discussion Section:

Add a brief note comparing your findings to similar published work (if available) to strengthen context.

Include more explicit acknowledgment of study limitations (e.g., sample size, potential selection bias).

Language Improvements:

In the abstract and results, avoid redundancy (e.g., “increase in levels was increased”).

Rephrase slightly awkward sentences in the conclusion to improve flow and clarity.

Figures and Tables:

Ensure all figure legends are self-explanatory.

Abbreviations should be defined at first use within tables and figures.

Once these minor issues are addressed, I believe the manuscript will be ready for publication.

**Do you want your identity to be public for this peer review?** For information about this choice, including consent withdrawal, please see our Privacy Policy

Reviewer #3: No

---

## [Author Response · Author response to Decision Letter 2]

23 Oct 2025

Academic Editor

PLOS One

Re: Manuscript ID /PONE-D-24-39264/

Diabetes management cascade in Ulaanbaatar, Mongolia

Dear Editors,

Thank you for the opportunity to revise our manuscript and for the thoughtful and

constructive comments provided by the reviewers. We appreciate the time and effort

invested in evaluating our work. We have carefully considered all the comments and

revised the manuscript accordingly. Below, we provide a detailed, point-by-point response

to each comment.

For clarity, we have included the reviewer/editor comments in bold, followed by our

responses in plain text. All changes made in the manuscript have been highlighted in the

revised version with tracked changes.

Reviewer Comments:

Comment: Add a brief note comparing your findings to similar published work (if

available) to strengthen context.

Response:

We have expanded the Discussion section to include a comparison of our results with

recent studies on diabetes prevalence and risk factors in similar populations. This

provides greater context and highlights consistencies and differences with previous

research.

Comment: Include more explicit acknowledgment of study limitations (e.g.,

sample size, potential selection bias).

Response:

We have added a dedicated paragraph in the Discussion section explicitly outlining the

study limitations, including the moderate sample size and potential selection bias due to

recruitment methods. These acknowledgments emphasize the need for cautious

interpretation and suggest directions for future research.

Comment: “Ensure all figure legends are self-explanatory.”

Response:

We have revised all figure legends to be fully self-explanatory. Each legend now clearly

describes the content of the figure, including sample sizes, variables measured, units,

and any abbreviations used, so that the figures can be understood independently of the

main text.

Comment: Abbreviations should be defined at first use within tables and figures.

Response:

We have defined all abbreviations at their first appearance within tables and figures,

either in the legend or footnotes, for clarity and reader convenience.

Comment: In the abstract and results, avoid redundancy (e.g., “increase in levels

was increased”).

Response:

We carefully reviewed the manuscript and rephrased redundant expressions to improve

clarity and conciseness, including in the Abstract and Results sections.

Sincerely Yours,

Naranjargal Dashdorj, Chief Executive Officer

Phone: +976 99199246

Email: dashdorj@onomfoundation.org

---

## [Editor Report · Decision Letter 2]

10 Nov 2025

Diabetes management cascade in Ulaanbaatar, Mongolia

PONE-D-24-39264R2

Dear Dr. Dashdorj,

We’re pleased to inform you that your manuscript has been judged scientifically suitable for publication and will be formally accepted for publication once it meets all outstanding technical requirements.

Kind regards,

Patricia Khashayar

Academic Editor

PLOS ONE
---

## [Editor Report · Acceptance letter]

PONE-D-24-39264R2

PLOS One

Dear Dr. Dashdorj,

I'm pleased to inform you that your manuscript has been deemed suitable for publication in PLOS One. Congratulations! Your manuscript is now being handed over to our production team.

Kind regards,

on behalf of

Dr. Patricia Khashayar

Academic Editor

PLOS One